# Green CURIOCITY: a study protocol for a European birth cohort study analysing childhood heat-related health impacts and protective effects of urban natural environments

Matilda van den Bosch [1,2,3] Xavier Basagaña,[1,2,3] Pierpaolo Mudu,[4] Vladimir Kendrovski,[4] Léa Maitre [1,2,3] Norun Hjertager Krog,[5] Gunn Marit Aasvang,[5] Regina Grazuleviciene,[6] Rosemary McEachan,[7] Martine Vrijheid,[1,2,3] Mark J Nieuwenhuijsen [1,2,3]

**To cite:** van den Bosch M, Basagaña X, Mudu P, *et al.* Green CURIOCITY: a study protocol for a European birth cohort study analysing childhood heat-related health impacts and protective effects of urban natural environments. *BMJ Open* 2022;**12**:e052537. doi:10.1136/bmjopen-2021-052537

For numbered affiliations see end of article.

**Correspondence to**
Dr Matilda van den Bosch;
matilda.vandenbosch@isglobal.org

## ABSTRACT

**Introduction** The European climate is getting warmer and the impact on childhood health and development is insufficiently understood. Equally, how heat-related health risks can be reduced through nature-based solutions, such as exposure to urban natural environments, is unknown. Green CURe In Outdoor CITY spaces (Green CURIOCITY) will analyse how heat exposure during pregnancy affects birth outcomes and how long-term heat exposure may influence children's neurodevelopment. We will also investigate if adverse effects can be mitigated by urban natural environments. A final goal is to visualise intraurban patterns of heat vulnerability and assist planning towards healthier cities.

**Methods and analysis** We will use existing data from the Human Early-Life Exposure cohort, which includes information on birth outcomes and neurodevelopment from six European birth cohorts. The cohort is linked to data on prenatal heat exposure and impact on birth outcomes will be analysed with logistic regression models, adjusting for air pollution and noise and sociobehavioural covariates. Similarly, impact of cumulative and immediate heat exposure on neurodevelopmental outcomes at age 5 will be assessed. For both analyses, the potentially moderating impact of natural environments will be quantified. For visualisation, Geographical information systems data will be combined to develop vulnerability maps, demonstrating urban 'hot spots' where the risk of negative impacts of heat is aggravated due to sociodemographic and land use patterns. Finally, geospatial and meteorological data will be used for informing GreenUr, an existing software prototype developed by the WHO Regional Office for Europe to quantify health impacts and augment policy tools for urban green space planning.

**Ethics and dissemination** The protocol was approved by the Comité Ético de Investigación Clínica Parc de Salut MAR, Spain. Findings will be published in peer-reviewed journals and presented at policy events. Through stakeholder engagement, the results will also reach user groups and practitioners.

### Strengths and limitations of this study

► The project will use harmonised data from a large pan-European cohort enabling assessments across different climate zones.

► The accessibility of long-term temporal exposure data facilitates analyses of vulnerability windows.

► Analyses of participants who have changed residence over the study period will partly address self-selection bias.

► The development of visualisation and policy tools in Green CURIOCITY will provide opportunities for immediate implementation and impact on healthy urban planning.

► Some of the cohort contains a disproportionately high number of participants from the highest educational level, potentially limiting the representativeness of the results.

## INTRODUCTION

The European climate is getting warmer and the summer of 2019 experienced multiple severe heat waves, shattering all-time temperature records in cities across the continent.[1] The morbidity and mortality toll of these events is yet to be unfolded, but we already know the dire effects on a number of vulnerable populations, including more than 20 children who became seriously ill after experiencing temperatures above 40°C in the Netherlands.[2] Following climate change, Europe can expect higher average daily temperatures and more frequent heatwaves[3] with severe health and economic consequences.[4] The forecast is even grimmer in cities than in rural areas due to the urban heat island effect[5] and long-term effects are insufficiently understood. Heat also aggravates negative

impacts of air pollution[6] and intensifies symptoms and mortality risk of many chronic diseases.[7 8] Many countries are unprepared to fully address this challenge, and some populations are at higher risks than others due to, for example, socioeconomic,[9] behavioural[10] or demographic factors[11] that reduce resilience. Understanding the full extent of climate change related health impacts across the life course is imperative to accurately act and address the issue, through mitigation, adaption and treatment options that are available for everyone.

Children are particularly vulnerable to both acute and long-term heat exposure due to a constellation of biological and behavioural factors, including physiology, metabolism and activity patterns.[12 13] This increases the risk of heat-related morbidity and negative impacts. Recent reviews and research[14 15] suggests that also prenatal heat exposure has negative effects, possibly because of an increase in oxidative stress and systemic inflammation in response to temperature changes during pregnancy.[16 17] However, additional evidence is clearly needed due to the paucity of studies that have examined the effects of long-term heat exposure on birth outcomes and childhood development. Prior research has largely focused on adults and little is known about how to reduce risks among children. To address this escalating threat, improved evidence and innovative solutions are urgently needed, including insights into risk differences due to sex, socioeconomic status (SES) and living context.

Nature-based solutions (NBS)—defined as 'actions which are inspired by, supported by or copied from nature' designed to address environmental challenges in an adaptive manner while providing economic, social and environmental benefits[18]—offer unique potential for addressing this threat. The European Union (EU) Research and Innovation Policy Agenda on NBS and Re-Naturing cities aims to position the EU as a leader in 'innovating with nature' for more resilient societies. In this context, urban natural environments (UNEs, comprising both green spaces such as parks and street trees and blue spaces such as rivers and lakes) stand out as effective NBS to reduce heat-related health risks by cooling via increased shade and evapotranspiration. Existing evidence suggests that the temperature-reducing effect of UNEs ranges from 0.4°C to 3.0°C, or even more, depending on local context,[19 20] with even-larger effects during the night.[21] Other, more technical, adaptation strategies exist, but many have proven to be less efficient[22] or energy-intensive (eg, air condition)[23] consequently escalating greenhouse gas emissions and exacerbating climate change.[24] The urgent need to identify more-sustainable solutions (including NBS) is recognised in a forthcoming *Lancet* series on Heat and Health.[25]

In parallel, an increasing amount of research indicates numerous childhood health benefits from UNEs, including reduced prevalence of asthma,[26] Attention Deficit Hyperactivity Disorder (ADHD),[27] and overweight,[28] and improved birth outcomes[29] and cognitive and other neurodevelopmental functions[30] with impact across the life course.[31] Various pathways for these health benefits have been suggested, but evidence is inconsistent. From this perspective, the mitigating effect of UNEs on heat and subsequent health impacts should be further investigated. As an example, UNEs may contribute to a generally cooler climate, reducing health-related harms of long-term heat exposure. A systematic review from 2016 concluded that the cooling effect of UNEs is one of the more-likely pathways between UNEs and health and that research on this mediator should be prioritised.[32]

## Knowledge gaps around climate change, childhood health and NBS

The full impact of the effects of climate change on children's health via acute and long-term heat exposure remains unclear, including differences between climate zones. To identify intraurban inequalities and areas of greater vulnerability, the roles of specific land-use and socioeconomic factors merit further investigation. The potential of NBS to prevent heat-related health risks must also be analysed in more depth. Although the elderly and chronically ill also represent vulnerable populations, the focus on children is critical from a Developmental Origin of Health and Disease perspective,[33] because prenatal and childhood exposures impact health across the life course. Cold-weather events also pose health risks,[34] but climate change is more clearly linked to heat-related health risks.[35] The research project Green CURe In Outdoor CITY spaces (Green CURIOCITY) will address these knowledge gaps and guide the implementation of sustainable and equitable solutions, including NBS, to address heat exposure and impaired childhood health and development. The topic is of high urgency and relevance because climate change will inevitably increase heat-related morbidity in vulnerable populations, including among preborn and children, and better understanding of NBS could be an approach to reduce current and future risks to health and society.

## Aims and objectives

The objective of Green CURIOCITY is to address the outlined knowledge gaps by analysing the association between prenatal and postnatal heat exposure and childhood birth and neurodevelopmental outcomes in different European climate zones. Further on, the aim is to identify whether neighbourhood exposure to UNEs is associated with a lower risk of heat-related adverse health outcomes among children. The goal is also to address implementation by identifying and mapping spatial 'hot spots' of childhood heat vulnerability in different European cities with the aim of visualising where interventions, such as NBS, should be prioritised. Finally, the project aims to influence policy-making by contributing to further information to a plugin tool, GreenUr, for geographical information system (GIS) software (QGIS), developed by the WHO Regional Office for Europe. GreenUr measures availability and accessibility of green space and quantifies direct and indirect health impacts

at a city scale and Green CURIOCITY aims to specifically contribute to opportunities for quantifying health benefits from urban green spaces within the context of heat reduction.

By the end of Green CURIOCITY, we expect to have identified negative associations between prenatal and postnatal heat exposure and childhood birth and neurodevelopmental outcomes. We also expect to find a lower risk of heat-related adverse health outcomes in areas where the abundance of urban natural spaces is high. These results, as well as the visualisation of 'hot spots', would have implications for future urban development and also spur research initiatives to address these issues on a global scale.

## METHODS AND ANALYSIS

The inferential studies of Green CURIOCITY are based on linkages of environmental exposure data to cohort data from the Human Early-Life Exposome (HELIX) project.[36] HELIX is a collaborative project of six ongoing, longitudinal, population-based birth cohort studies in Europe funded through the EU FP7 Exposome Research programme. Pregnant women in the original cohorts were recruited between 1999 and 2010, and all cohorts included at least one follow-up during the prenatal period, at birth and after delivery. The complete HELIX study population includes 31 472 mother–child pairs at the time of delivery and all information is stored in a common data warehouse. The harmonisation process of data from the national cohorts in HELIX followed expert input for standardisation rules and was based on a coding system, subsequent to checking and matching of cohort-specific variables.[36]

### Study area and national cohorts

The birth cohorts of HELIX include the Born in Bradford study (BiB) in the UK,[37] the Étude des Déterminants pré et postnatals du développement et de la santé de l'Enfant study (EDEN) in France,[38] the INfancia y Medio Ambiente cohort (INMA) in Spain,[39] the Kaunas cohort (KANC) in Lithuania,[40] the Norwegian Mother, Father

and Child Cohort Study (MoBa)[41] and the Mother Child Cohort study (RHEA) in Crete, Greece[42] (table 1).

Primary outcomes in Green CURIOCITY: Birth outcome data are available for the entire cohort (n=31 472) and include gestational duration, birth weight and length, and size for gestational age. Exclusion criteria in most of the cohorts were twin pregnancies, women with diabetes or pregnancy induced hypertension, and mothers under 16 years of age.

The availability of neurodevelopmental indicators varies across the cohorts, but data have been harmonised for cognitive function (verbal and non-verbal) and for psychomotor function (fine and gross motor function). In total 11 423 children have harmonised data for cognitive and/or psychomotor function linked to environmental exposures. Not all cohorts collected data for all the different outcomes (table 2). KANC did not collect data on neurodevelopmental outcomes.

All outcome variables have been harmonised and standardised to a mean of 100 and an SD of 15, making them comparable across the cohorts. The tests that were used for various neurodevelopmental outcomes in the respective cohorts are listed in table 3. All tests were evaluated at the age of 4.5 years.

### Covariates

The interactions between environmental exposures, childhood health outcomes and other social, biological and cultural variables are complex and the evidence of associations is often inconsistent. The final selection of confounder variables to include in the analyses will be determined a priori based on existing evidence and by using a Directed Acyclic Graphs approach. HELIX contains extensive information on a number of key covariates and based on a preliminary literature search on potential confounders of the association between prenatal heat exposure and birth outcomes, we will consider including child sex, season of birth,[43 44] urban living during pregnancy,[45] maternal age and education,[46 47] maternal body mass index,[48] parental country of origin,[49] maternal health behaviour (such as smoking and

| Table 1 | Summary of the national cohorts included in HELIX, adapted from Maitre *et al*[36] | | |
|---|---|---|---|
| **Cohort** | **Years of birth** | **Region covered by HELIX** | **No of births in HELIX entire cohort** |
| BiB, UK | 2007–2010 | Bradford | 10 849 |
| EDEN, France | 2003–2006 | Nancy and Poitiers, urban areas | 1900 |
| INMA, Spain | 2003–2008 | Gipuzkoa, Sabadell, Valencia | 2063 |
| KANC, Lithuania | 2007–2008 | Kaunas | 4107 |
| MoBa, Norway | 1999–2008 | Oslo | 11 095 |
| RHEA, Greece | 2007–2008 | Heraklion | 1458 |
| Total | | | 31 472 |

BiB, Born in Bradford; EDEN, Étude des Déterminants pré et postnatals du development et de la santé de l'Enfant; HELIX, Human Early-Life Exposome; INMA, INfancia y Medio Ambiente; KANC, Kaunas cohort; MoBa, Norwegian Mother, Father and Child Cohort Study; RHEA, Mother Child Cohort Study.

**Table 2** Harmonised neurodevelopmental outcome data per cohort to be used in green CURIOCITY, (N=11 423)

| Neurodevelopmental outcome | BiB | EDEN | INMA | MoBa | RHEA | Subjects in harmonised dataset |
|---|---|---|---|---|---|---|
| Cognition function | 0 | 1100 | 1420 | 0 | 779 | 3299 |
| Verbal cognitive function | 2057 | 1107 | 1420 | 5834 | 779 | 11 197 |
| Non-verbal cognitive function | 0 | 1107 | 1420 | 0 | 779 | 3306 |
| Psychomotor function | 0 | 0 | 1420 | 5939 | 779 | 8138 |
| Gross motor function | 0 | 0 | 1420 | 5864 | 779 | 8112 |
| Fine motor function | 1909 | 1120 | 1420 | 5913 | 779 | 11 092 |

BiB, Born in Bradford; EDEN, Étude des Déterminants pré et postnatals du dévelopment et de la santé de l'Enfant; INMA, INfancia y Medio Ambiente; MoBa, Norwegian Mother, Father and Child Cohort Study; RHEA, Mother Child Cohort Study.

alcohol intake) during pregnancy,[50] and individual and area level socioeconomic deprivation indicators.[48] The analysis of neurodevelopmental outcomes will consider the same confounders and, in addition, maternal cognitive function[51] and attendance to nursery school.[52]

### Environmental exposure variables

Environmental exposure data for Green CURIOCITY exist in an integrated GIS environment, contained in the HELIX database. Using complete residential address histories, exposure estimates per trimester during the prenatal period and average annual values up to age of neurodevelopmental assessment have been assigned to the participants.

By the time of submission and approval of this proposal, a heat index was planned to be developed based on data from local meteorological stations contained within the HELIX GIS-environment, including[1] temperature (average of mean/max/min during pregnancy period, per trimester and at birth)[3]; relative humidity (average of mean/max/min during pregnancy period, per trimester and at birth) and[4] atmospheric pressure during pregnancy and per trimester. The aim was to also include land surface temperature. Based on development of the HELIX GIS-environment, this project now aims to use a heat index derived from the E-OBS dataset within the Copernicus Programme (https://climate.copernicus.

**Table 3** The respective tests and scales used in the separate cohorts before harmonisation

| | Cognitive function | | | Psychomotor function | | |
|---|---|---|---|---|---|---|
| | Cognitive function | Verbal cognition | Non-verbal cognition | Psychomotor function | Gross motor | Fine motor |
| **BiB, UK** | | | | | | |
| British Picture Vocabulary Scale[66] | | x | | | | |
| Clinical Kinematic Assessment Tool[67] | | | | | | x |
| **EDEN, France** | | | | | | |
| Weschler Preschool and Primary Scale of Intelligence[68] | x | x | x | | | |
| Peg moving task[69] | | | | | | x |
| **INMA, Spain** | | | | | | |
| McCarthy Scales of Children's Abilities[70] | X | X | X | X | X | X |
| **MoBa, Norway** | | | | | | |
| Child Development Inventory[71] | | X | | X | X | X |
| **RHEA, Greece** | | | | | | |
| McCarthy Scales of Children's Abilities[70] | x | x | x | x | x | x |

BiB, Born in Bradford; EDEN, Étude des Déterminants pré et postnatals du dévelopment et de la santé de l'Enfant; INMA, INfancia y Medio Ambiente; RHEA, Mother Child Cohort Study.

van den Bosch M, *et al. BMJ Open* 2022;**12**:e052537. doi:10.1136/bmjopen-2021-052537

eu/). This index is derived using daily minimum and maximum temperature, daily precipitation sum, global radiation, and relative humidity and the spatial resolution is 0.1° (approximately 11 km). The values are calculated annually, monthly and seasonally. To address specific time windows of potential vulnerability, we will also test, in separate models, impact of temperature, relative humidity and atmospheric pressure, using the time-specific data contained in the HELIX dataset. The meteorological values per trimester will be linked to data on birth outcomes and the opportunity for assessing specific heat waves will be explored. Impact on neurodevelopment will be assessed by linking the heat index value prenatally as well as postnatal annual values up to the age of outcome assessment.

To assess the modifying impact of urban natural spaces on the association between heat exposure and childhood health outcomes, we will use data from the European Environmental Agency's 'Urban Atlas' (year 2006),[53] apart from for the Norwegian cohort (MoBa) where a similar local land cover map, Norwegian map, N50, was used.[54] The Urban Atlas contains 17 land use classes and is developed from image classification and interpretation of high-resolution satellite imagery. The accuracy level is approximately 85% for artificial surfaces. Across European cities, it provides harmonised information of land use, including, for example, green urban areas, water, sport and recreational facilities, impervious surfaces and built-up areas. We assume relatively minor changes in urban green space land cover over the study period. Apart from exposure values, we will use area and distance to nearest green and blue spaces, as of previous protocols.[55 56] We will also use a general vegetation indicator, the Normalised Difference Vegetation Index (NDVI),[57] derived from 30 m spatial resolution Landsat images and provided in raster maps. To achieve maximum exposure contrast, the NDVI values that are included in the HELIX GIS-environment are based on available cloud-free Landsat TM images during the period between May and August for years relevant to our period of study. Greenness values are calculated and averaged within 100, 300 and 500 m buffers around each residential address. These buffer zones are commonly used in epidemiological studies assessing the impact of natural spaces on health outcomes. In general, the 300 m buffer zone is considered as most indicative of home exposure,[58] but additional buffer zones will also be tested for sensitivity analyses. The project will adjust for air pollution exposure ($NO_2$, NOx, $PM_{10}$, $PM_{2.5}$) available from existing land use regression (LUR) models developed in the European Study of Cohorts for Air Pollution Effects project. See Beelen et al[59] and Eeftens et al[60] for further information about the LUR models. The values are extrapolated back in time, using data from existing regulatory monitors, and averaged per trimester, pregnancy period, birth and annually. In addition, we will control for noise exposure, available from European road traffic noise maps from the European Environment Agency with continuous and categorical values ($L_{den}$).

## Statistical analysis

We will use logistic regression to estimate the association between trimester-specific temperature and risk of short gestational duration or small for gestational age and linear regression to estimate the association between trimester-specific temperature and term birth weight, standardised via z-scores and adjusted for key covariates. Results will be pooled overall and by geographic regions and climate zones (Mediterranean: Spain and Greece; Temperate: UK and France; Boreal: Norway). The pooling per climate zone will contribute to the understanding of potential differences in the adaptation to heat as well as in the modifying effect of natural spaces on the selected outcomes. In addition, stratified analyses will be conducted to assess moderation by sex and SES. We will develop structural equation models (SEMs) to explore the potential for UNEs to mitigate adverse heat-related effects on birth outcomes. SEM analyses will allow us to test the statistical significance of a mediation effect (ie, that improved birth outcomes occur via cooling effects of UNEs), while adjusting for confounding and neighbourhood dependency. Using SEM will also allow us to detect indirect effects (ie, that UNEs reduce harmful effects of heat even if direct heat-related effects on birth outcomes are not detected). Our primary approach will be to use a causal mediation analysis framework, as suggested by Imai et al,[61] but a detailed statistical plan will be developed as the project proceeds and alternative approaches, such as partial least squares-SEMs[62] will also be considered. Results will be pooled by climate zone and we will run sensitivity analyses adjusting for air pollution and noise. We will also develop logistic regression models to test the association between long-term heat exposure and neurodevelopment, adjusting for key confounders. We will apply time-weighted averages across the full exposure period, prenatally and postnatally, carrying out subanalyses among children who have moved to assess the impact of exposure length and to partially address self-selection bias. Additional analyses will be stratified by sex and SES and pooled by geographic region and climate zone. We will use SEM with a similar approach and rationale as for the birth outcome analyses, assessing the mitigating effect of UNE exposure, as well as similar sensitivity analyses.

## Mapping and spatial overlay design

For the 'visualisation' study of Green CURIOCITY we will conduct a spatial overlay analysis to identify areas where different environmental (eg, low amount of UNEs) or demographic (eg, low age or low income) vulnerability factors coincide with high temperatures, so called urban 'hot spots'. We will assess the approach and consider different methods based on spatial clustering, such as LISA or multivariate clustering.[63 64] Geospatial data capturing the intraurban spatial distribution of environmental (heat, UNE, built environment and impervious surfaces) vulnerability indicators will be downloaded from the HELIX GIS environment and demographic data (age group, sex and SES) from municipal resources.

Environmental data will be averaged per the spatial unit that is aligned with SES data aggregation (eg, dissemination areas) in respective cities. We primarily aim to use a simple combination overlay analysis, but more complex methods, such as union, intersection or identity operations may also be considered. Subsequently, we will apply a ranking method and identify the most-vulnerable areas, based on prevalence of vulnerability factors, using a cut-off of 25%, similar to the approach used by Aminipouri *et al.*[65] The 25% threshold is an easily interpretable number, which makes the approach useful for city officials to replicate for making decisions about directing investments in NBS interventions.

## Software expansion

For the implementation tool, expansion of an existing software prototype developed for QGIS by WHO (GreenUr, https://www.euro.who.int/en/health-topics/environment-and-health/urban-health/activities/greenur-the-green-urban-spaces-and-health-tool) will be developed based on adaptation of existing algorithms and modelled quantification of heat related health effects following changes in UNEs. Raster (Landsat) and vector data (Urban Atlas) of UNEs in combination with meteorological input data will be incorporated in the models. GreenUr is a prototype under development and the exact input data and what functions to incorporate will depend on the general evolvement of Green CURIOCITY and on the current processing of the prototype.

## Patient and public involvement

Neither patients nor the public were involved in the development of this protocol. Patient and public involvement were, however, part of the wider HELIX project per each cohort.

## ETHICS AND DISSEMINATION

All data to be used in Green CURIOCITY are available from the HELIX cohort. The six national cohorts on which HELIX are based had undergone ethical review and approval at initiation and new approvals were acquired for the work in HELIX. An additional evaluation and ethics approval has been acquired for Green CUROICTY project from the Comité Ético de investigación Clínica Parc de Salut MAR, Spain (reference number 2020/9567/I). The data are pseudonymised and all procedures of Green CURIOCITY that will include analyses of study participants will follow the ethics guidelines of ISGlobal's Ethics Committee and the HELIX Ethics Task Force.[36] Data will be safeguarded to protect the confidentiality and anonymity of the participants and is consistent with the terms of the consent signed by participants in the respective national cohorts. Adequate measures to ensure health data protection and confidentiality will be taken, according to European directive on the protection of individuals with regard to the processing of personal data and free movement of such

data (2016/679, General Data Protection Regulation). All reported results will pertain to analyses of aggregate data; no variables or combination of variables that can identify an individual will be associated with any published or unpublished report of the study.

Results from Green CURIOCITY will be disseminated through high-impact scientific journals, conference presentations, expert seminars and policy briefs. The project has connections to broad international, interdisciplinary networks, such as the International Union for Conservation of Nature and the WHO, which will support dissemination to a global audience. Methods and results will also be communicated through research briefs and reports and via institutional social media platforms and project websites. There are plans for roundtable events with urban planners and key decision-makers to share results of the vulnerability-mapping exercises and to integrate findings into sustainable urban plans, although these strategies may have to be adapted due to the current COVID-19 situation. In addition, the project aims to prepare educational material for distribution at schools, healthcare clinics and public health websites, including infographics and worksheets describing how heat impacts children, summarising recommendations regarding 'green' schoolyards and playgrounds, and highlighting the importance of heat prevention during pregnancy.

**Author affiliations**

[1]Air pollution and Urban Environment, Barcelona Institute for Global Health, Barcelona, Spain
[2]Universitat Pompeu Fabra, Barcelona, Spain
[3]CIBER Epidemiología y Salud Pública (CIBERESP), Madrid, Spain
[4]World Health Organization European Centre for Environment and Health, Bonn, Nordrhein-Westfalen, Germany
[5]Air Quality and Noise, Norwegian Institute of Public Health, Oslo, Norway
[6]Department of Environmental Sciences, Vytauto Didziojo Universitetas, Kaunas, Lithuania
[7]Bradford Institute for Health Research, Bradford, UK

**Contributors** MvdB drafted this manuscript on the basis of a grant proposal that was devised and written by MvdB in collaboration with MJN, XB, PM and VK. MV is principal investigator of HELIX. LM coordinated the collection and harmonisation of the entire HELIX database. GMA, RG and RM coordinated HELIX data collection and research protocols in respective national cohorts. NHK provided environmental data for MoBa. All authors contributed to drafting, editing, and reviewing the manuscript. All authors read the final version of this manuscript, approved it for submission for publication, and agreed to be accountable for all aspects of the work. The proposal was approved for funding on 31 March 2020.

**Funding** This work is supported by the European Commission through a Horizon 2020 Marie Skłodowska-Curie Grant, Reference number 891 538.LM is funded by a Juan de la Cierva-Incorporación fellowship (IJC2018-035394-I) awarded by the Spanish Ministerio de Economía, Industria y Competitividad.

**Competing interests** None declared.

**Patient and public involvement** Patients and/or the public were involved in the design, or conduct, or reporting, or dissemination plans of this research. Refer to the Methods section for further details.

**Patient consent for publication** Not applicable.

**Provenance and peer review** Not commissioned; externally peer reviewed.

**ORCID iDs**

Matilda van den Bosch http://orcid.org/0000-0003-1410-0099

Léa Maitre http://orcid.org/0000-0003-3682-7117

Mark J Nieuwenhuijsen http://orcid.org/0000-0001-9461-7981

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
