## [Reviewer comments · BMJ Open]

ARTICLE DETAILS

TITLE (PROVISIONAL)	Green CURIOCITY: A study protocol for a European birth cohort study analysing childhood heat-related health impacts and protective effects of urban natural environments.
AUTHORS	van den Bosch, Matilda; Basagaña, Xavier; Mudu, Pierpaolo; Kendrovski, Vladimir; Maitre, Léa; Hjertager Krog, Norun; Aasvang, Gunn Marit; Grazuleviciene, Regina; McEachan, Rosemary; Vrijheid, Martine; Nieuwenhuijsen, Mark

VERSION 1 – REVIEW

REVIEWER	Fry, Richard Swansea University, Medical School
REVIEW RETURNED	30-Jun-2021

GENERAL COMMENTS	This sounds like a really interesting study and I look forward to the results as they come out. I have a couple of comments for the authors: p10. You specify 100,300 and 500m as your buffers zones for modelling exposures. Are these buffers for the NDVI calculations only, or are they for all environmental exposure variables within this study? Are there any a priori justifications for the distance buffers from an epidemiological perspective that can be included - or is it more of a 'data mining' approach to see where the strongest associations lie. Either way it would be good to clarify why these buffers have been chosen. Generally, it would be good to expand on the 'pooling by climate zones'. Obviously, there are differences across Europe in terms of climate but it is not clear what the different climate zones of interest are and what the hypothesis is for each zone (for example what is the expected health impacts of damper more humid winters in Northern Europe) in relation to climate change. Setting these high level questions out in the protocol would enhance readers understanding of the project objectives.
--

REVIEWER	Liu, Tao Guangdong Provincial Institute of Public Health
REVIEW RETURNED	29-Aug-2021

GENERAL COMMENTS	Thanks for giving me an opportunity to review this study protocol which aims to identify whether neighbourhood exposure to UNEs is associated with a lower risk of heat-related adverse health
--

	outcomes among children, and to identify and map the spatial “hotspots” of childhood heat vulnerability in different European cities. This study is based on the HELIX project which is a collaborative project of six ongoing, longitudinal, population-based birth cohort studies in Europe funded through the EU FP7 Exposome Research programme. Generally, this is a well written manuscript. I have several issues the authors need to concern:  1. This protocol aims to use data from six cohorts which covered different duration and regions. My major concern is the different definitions of exposures and outcomes, which may lead to bias to their findings. For example, methods of defining the neurodevelopmental outcome are substantially different across those cohort studies, which may lead to a large heterogeneity to their individual results. 2. The roadmap to assess the heat exposure is not clear. For example, what are the spatio-temporal resolutions of these meteorological variables? How to combine those four meteorological variables? Therefore, more detailed methods are needed in the manuscript. 3. What are the temporal resolutions of Landsat images, daily, weekly, or monthly? 4. The authors should give more information of the air pollution exposure assessment. 5. The authors need to integrate the results of modification analyses into the hotspots analysis. 6. Would you please briefly describe the expected results, and their implications?
--	--

REVIEWER	Labib, SM Cambridge University, MRC Epidemiology unit
REVIEW RETURNED	30-Aug-2021

GENERAL COMMENTS	Thanks for inviting me to review Green CURIOCITY protocol. I was happy to read the protocol in advance. Using the HELIX cohorts, this new study would explore the impact of heat exposure on birth outcomes and neurodevelopment while taking into account the mediating or moderating effects of urban natural environments. The protocol is detailed and links very well with other HELIX protocols. And generally, it provided the necessary information regarding the data sources, time, and analytical approaches planned. However, I have a few minor comments on clarifying several points and suggested some potential additional methodological approaches to consider.  1. Heat exposure index (Page 7, line 9): There are four variables for this, LST is spatially explicit and local, where others are non-spatial and from the weather station. How would these be harmonized- such as are these be harmonized at each residential location point or over a certain area (e.g., city or neighborhood)? Also, I wonder, will this index be a continuous score or a categorical index? A bit more details might help. 2. More details on the LST are warranted (Page 7, line 10-11), such as were the estimates consider the highest LST in a month or composite LST over the months? Maybe there are also opportunities to explore the seasonality effects (winter vs. summer month births) if the HELIX data has monthly data. Maybe also possible to estimate LST from Landsat for those months to explore both seasonality and heatwaves. One thing to note, the LST in HELIX data is only for 50 m buffer (Maitre et al., 2018). I wonder, is
--

this sufficient to capture the effects on the surrounding built and natural environment that may influence the LST itself, are there opportunities to check other buffers? Maybe a point to consider when analyzing the data.

3. Urban Atlas data period (Page 7, line 22): I wonder how the Urban Atlas data of three periods (2006, 2012, and 2018) matched with the cohort timelines. From the data collection period, it appeared only 2006 urban atlas data had been used for these cohorts. Maybe adding information about the map year of Urban Atlas will clarify.

4. Structural equation models (Page 7, line 48) may need a bit more on what type of SEM would be used. There are two schools of thought on SEM, one is focused on modeling existing established theory (covariance-based model/CB-SEM), and another is for predicting new pathways (PLS-SEM model). CB-SEM is primarily used to confirm (or reject) theories. PLS-SEM is primarily used to develop theories in exploratory research. It would be better to flash out which structure would be adopted in these studies. I think the PLS-SEM could provide new insights on moderation and mediation for new pathways. But I can see that the CB-SEM model can also explain the mediations. Please check these for details:
Hair Jr, J.F., Hult, G.T.M., Ringle, C.M. and Sarstedt, M., 2021. A primer on partial least squares structural equation modeling (PLS-SEM). Sage publications. (Chapter 1)
Astrachan, C.B., Patel, V.K. and Wanzanried, G., 2014. A comparative study of CB-SEM and PLS-SEM for theory development in family firm research. *Journal of Family Business Strategy*, 5(1), pp.116-128. doi: <https://doi.org/10.1016/j.jfbs.2013.12.002>

5. Spatial overlay analysis (page 8, line 11), while the overlay would identify the area-based variables coinciding with temperature, I believe this visualization can be future developed based on spatial clustering methods such as LISA clustering or multivariate clustering. These are better for hotspot identification. Such methods will identify and visualize the spatial pattern of high-low clusters in temperature with other variables such as high-low UNEs and high deprived-low deprived demographic regions. I also think the authors could clarify a bit more on what type of overlay (e.g., identity, intersect) they like to use. Please check the link for LISA: https://geodacenter.github.io/workbook/6a_local_auto/lab6a.html

6. Software development (Page 9, line 27): I am really excited to learn about this tool. I think the authors might want to discuss a little more how the results would be incorporated in GreenUr. Such as will there be an additional function to use the modeled relations from this study to estimate health impact from LST raster maps? Will it be possible to draw dose-response function from GreenUr? What meteorological inputs would be required and which model (line 36)?

7. Testing different spatial units (page 7, line 8): It would be good to provide some details on the spatial units. How were these defined, and how can they be sensitive? Are these different buffer sizes, admin boundaries? As I found in the HELIX cohort paper indicated 100, 300, and 500 m buffer?

As a whole, this is an exciting and much-needed longitudinal study! I am looking forward to seeing the results of this study.

VERSION 1 – AUTHOR RESPONSE

Reviewer: 1

Dr. Richard Fry, Swansea University

Comments to the Author:

This sounds like a really interesting study and I look forward to the results as they come out. I have a couple of comments for the authors:

- Thank you very much for your encouraging words and for your constructive comments.

p10. You specify 100,300 and 500m as your buffers zones for modelling exposures. Are these buffers for the NDVI calculations only, or are they for all environmental exposure variables within this study?

Are there any a priori justifications for the distance buffers from an epidemiological perspective that can be included - or is it more of a 'data mining' approach to see where the strongest associations lie. Either way it would be good to clarify why these buffers have been chosen.

- We selected these buffer zones to make it comparable with most previous studies on the topic of natural spaces and health outcomes. In general, the 300m buffer zone seem to be most indicative of potential health associations, but other distances have also been used for sensitivity analyses. It should also be noted that the green space values tend to be highly correlated across the buffer zones. We added this text for clarification in the manuscript (line 263 – 266):

“These buffer zones are commonly used in epidemiological studies assessing the impact of natural spaces on health outcomes. In general, the 300m buffer zone is considered as most indicative of home exposure (64), but additional buffer zones will also be tested for sensitivity analyses”:

Generally, it would be good to expand on the 'pooling by climate zones'. Obviously, there are differences across Europe in terms of climate but it is not clear what the different climate zones of interest are and what the hypothesis is for each zone (for example what is the expected health impacts of damper more humid winters in Northern Europe) in relation to climate change. Setting these high level questions out in the protocol would enhance readers understanding of the project objectives.

- We have clarified what the different climate zones are and what cohorts/countries belong to each and also explained the rational behind the assessment per climate zone (line 280 - 283):

“Results will be pooled overall and by geographic regions and climate zones (Mediterranean: Spain and Greece; Temperate: UK and France; Boreal: Norway). The pooling per climate zone will contribute to the understanding of potential differences in the adaptation to heat as well as in the modifying effect of natural spaces on the selected outcomes.”

Reviewer: 2

Dr. Tao Liu, Guangdong Provincial Institute of Public Health

Comments to the Author:

Thanks for giving me an opportunity to review this study protocol which aims to identify whether neighbourhood exposure to UNEs is associated with a lower risk of heat-related adverse health outcomes among children, and to identify and map the spatial “hotspots” of childhood heat vulnerability in different European cities. This study is based on the HELIX project which is a collaborative project of six ongoing, longitudinal, population-based birth cohort studies in Europe

funded through the EU FP7 Exposome Research programme. Generally, this is a well written manuscript. I have several issues the authors need to concern:

- Thank you for your time in reviewing our study protocol and for providing several constructive suggestions.

1. This protocol aims to use data from six cohorts which covered different duration and regions. My major concern is the different definitions of exposures and outcomes, which may lead to bias to their findings. For example, methods of defining the neurodevelopmental outcome are substantially different across those cohort studies, which may lead to a large heterogeneity to their individual results.

- As of the manuscript, the outcome data are harmonized across the cohorts:

Line 195-198: "The availability of neurodevelopmental indicators varies across the cohorts, but data have been harmonised for cognitive function (verbal and non-verbal) and for psychomotor function (fine and gross motor function). In total 11,423 children have harmonised data for cognitive and/or psychomotor function linked to environmental exposures."

Line 203 - 204: "All outcome variables have been harmonised and standardised to a mean of 100 and a standard deviation of 15, making them comparable across the cohorts."

Line 178 – 180: "The harmonisation process of data from the national cohorts in HELIX followed expert input for standardisation rules and was based on a coding-system, subsequent to checking and matching of cohort-specific variables (36)."

Regarding the environmental exposure data, they are all integrated in the HELIX

GIS environment and standardised across the study areas, see line 225-226 in manuscript:

"Environmental exposure data for Green CURIORITY are standardised and exist in an integrated Geographic Information System (GIS) environment, contained in the HELIX database."

The environmental exposure measures are derived from European exposure models, see line 249 regarding natural space (from Urban Atlas and a similar Norwegian equivalent), line 266-269 regarding air pollution (from the ESCAPE project as of reference, 65 and 66, and line 272-273 regarding noise (European road traffic noise map). For the noise values, we have now added in the text (line 272-273) that they are acquired from the European Environment Agency.

2. The roadmap to assess the heat exposure is not clear. For example, what are the spatio-temporal resolutions of these meteorological variables? How to combine those four meteorological variables? Therefore, more detailed methods are needed in the manuscript.

- Since the proposal was submitted and approved for funding by the European Commission within the Marie Skłodowska-Curie Actions funding scheme, a new approach for measuring heat within the HELIX cohort will be implemented, based on climate indices derived from the E-OBS dataset within the Copernicus Programme (<https://climate.copernicus.eu/>). The text in the manuscript has now been updated to reflect this change (line 230-243):

"By the time of submission and approval of this proposal, a heat index was planned to be developed based on data from local meteorological stations contained within the HELIX GIS-environment, including (1) temperature (average of mean/max/min during pregnancy period, per trimester, and at birth); (3) relative humidity (average of mean/max/min during pregnancy period, per trimester, and at birth); and (4) atmospheric pressure during pregnancy and per trimester. In addition, the aim was to include land surface temperature (LST). Based on development of the HELIX GIS-environment, this project now aims to use a heat index derived from the E-OBS dataset within the Copernicus Programme (<https://climate.copernicus.eu/>). This index is derived using daily minimum and maximum temperature, daily precipitation sum, global radiation, and relative humidity. The values are calculated annually, monthly, and seasonally. To address specific time windows of potential vulnerability, we will also test, in separate models, impact of temperature, relative humidity, and atmospheric pressure using the time specific data contained in the HELIX dataset. The meteorological values per

trimester will be linked to data on birth outcomes and the opportunity for assessing specific heat waves will be explored. Impact on neurodevelopment will be assessed by linking the heat index value prenatally as well as postnatal annual values up to the age of outcome assessment.”

3. What are the temporal resolutions of Landsat images, daily, weekly, or monthly?

We added this text in the manuscript to clarify (line 259-266):

“To achieve maximum exposure contrast, the NDVI values that are included in the HELIX GIS-environment are based on available cloud-free Landsat TM images during the period between May and August for years relevant to our period of study. Greenness values are calculated and averaged within 100-, 300-, and 500-m buffers around each residential address. These buffer zones are commonly used in epidemiological studies assessing the impact of natural spaces on health outcomes. In general, the 300m buffer zone is considered as most indicative of home exposure (64), but additional buffer zones will also be tested for sensitivity analyses.”

4. The authors should give more information of the air pollution exposure assessment.

- In this project, we will only include air pollution variables as covariates in the models and we therefore refer the reader to review previous publications (Beelen et al. 2014: <https://www.sciencedirect.com/science/article/pii/S1352231013001386> and Eeftens et al. 2012: <https://pubs.acs.org/doi/abs/10.1021/es301948k>) to gain more insight around the development of the LUR models. We have adjusted the text (line 267 – 269):

“The project will adjust for air pollution exposure (NO₂, NO_x, PM₁₀, PM_{2.5}) available from existing land use regression (LUR) models developed in the European Study of Cohorts for Air Pollution Effects (ESCAPE) project. See Beelen et al. (65) and Eeftens et al. (66) for further information about the LUR models.”

5. The authors need to integrate the results of modification analyses into the hotspots analysis.

- This is an interesting idea that we will consider as the project proceeds. However, we cannot add this to the current manuscript, since it is based on what has been approved by reviewers in European Commission panel for the Marie Skłodowska-Curie Actions funding scheme. We can therefore not make any content changes to the protocol.

6. Would you please briefly describe the expected results, and their implications?

- Although we cannot make too much changes regarding assumptions compared to the text that was approved by the EU funding agency, we have added this section to the Aims and Objectives paragraph (line 162-167):

“By the end of Green CURIORITY, we expect to have identified negative associations between pre- and postnatal heat exposure and childhood birth and neurodevelopmental outcomes. We also expect to find a lower risk of heat-related adverse health outcomes in areas where the abundance of urban natural spaces is high. These results, as well as the visualisation of “hotspots”, would have implications for future urban development and also spur research initiatives to address these issues on a global scale.”

Reviewer: 3

Dr. SM Labib, Cambridge University

Comments to the Author:

Thanks for inviting me to review Green CURIORITY protocol. I was happy to read the protocol in advance. Using the HELIX cohorts, this new study would explore the impact of heat exposure on birth outcomes and neurodevelopment while taking into account the mediating or moderating effects of urban natural environments. The protocol is detailed and links very well with other HELIX protocols. And generally, it provided the necessary information regarding the data sources, time, and analytical approaches planned. However, I have a few minor comments on clarifying several points and suggested some potential additional methodological approaches to consider.

- Thank you for your encouraging words and for your useful comments and suggestions.

1. Heat exposure index (Page 7, line 9): There are four variables for this, LST is spatially explicit and local, where others are non-spatial and from the weather station. How would these be harmonized—such as are these be harmonized at each residential location point or over a certain area (e.g., city or neighborhood)? Also, I wonder, will this index be a continuous score or a categorical index? A bit more details might help.

- Since the proposal was submitted and approved for funding by the European Commission within the Marie Skłodowska-Curie Actions funding scheme, a new approach for measuring heat within the HELIX cohort will be implemented, based on climate indices derived from the E-OBS dataset within the Copernicus Programme (<https://climate.copernicus.eu/>). The text in the manuscript has now been updated to reflect this change (line 230-243):

“By the time of submission and approval of this proposal, a heat index was planned to be developed based on data from local meteorological stations contained within the HELIX GIS-environment, including (1) temperature (average of mean/max/min during pregnancy period, per trimester, and at birth); (2) relative humidity (average of mean/max/min during pregnancy period, per trimester, and at birth); and (3) atmospheric pressure during pregnancy and per trimester. In addition, the aim was to include land surface temperature (LST). Based on development of the HELIX GIS-environment, this project now aims to use a heat index derived from the E-OBS dataset within the Copernicus Programme (<https://climate.copernicus.eu/>). This index is derived using daily minimum and maximum temperature, daily precipitation sum, global radiation, and relative humidity and the resolution is 0.1° (approximately 11 km). The values are calculated annually, monthly, and seasonally. The meteorological values per trimester will be linked to data on birth outcomes and the opportunity for assessing specific heat waves will be explored. Impact on neurodevelopment will be assessed by linking the heat index value prenatally as well as postnatal annual values up to the age of outcome assessment.”

2. More details on the LST are warranted (Page 7, line 10-11), such as were the estimates consider the highest LST in a month or composite LST over the months? Maybe there are also opportunities to explore the seasonality effects (winter vs. summer month births) if the HELIX data has monthly data. Maybe also possible to estimate LST from Landsat for those months to explore both seasonality and heatwaves. One thing to note, the LST in HELIX data is only for 50 m buffer (Maitre et al., 2018). I wonder, is this sufficient to capture the effects on the surrounding built and natural environment that may influence the LST itself, are there opportunities to check other buffers? Maybe a point to consider when analyzing the data.

- As of above, we will initially focus on a different heat index than what was originally mentioned in the protocol. However, we appreciate the suggestion for additional analyses, that we think will be very interesting to explore and definitely something to consider as the project proceeds.

3. Urban Atlas data period (Page 7, line 22): I wonder how the Urban Atlas data of three periods (2006, 2012, and 2018) matched with the cohort timelines. From the data collection period, it

appeared only 2006 urban atlas data had been used for these cohorts. Maybe adding information about the map year of Urban Atlas will clarify.

- The land cover data that are linked to the cohorts are from Urban Atlas 2006. This has now been added to the manuscript (line 249). In addition, this sentence was inserted (line 255-256) to justify the temporal mismatch:

“We assume relatively minor changes in urban green space land cover over the study period.”

4. Structural equation models (Page 7, line 48) may need a bit more on what type of SEM would be used. There are two schools of thought on SEM, one is focused on modeling existing established theory (covariance-based model/CB-SEM), and another is for predicting new pathways (PLS-SEM model). CB-SEM is primarily used to confirm (or reject) theories. PLS-SEM is primarily used to develop theories in exploratory research. It would be better to flash out which structure would be adopted in these studies. I think the PLS-SEM could provide new insights on moderation and mediation for new pathways. But I can see that the CB-SEM model can also explain the mediations. Please check these for details:

Hair Jr, J.F., Hult, G.T.M., Ringle, C.M. and Sarstedt, M., 2021. A primer on partial least squares structural equation modeling (PLS-SEM). Sage publications. (Chapter 1)

Astrachan, C.B., Patel, V.K. and Wanzanried, G., 2014. A comparative study of CB-SEM and PLS-SEM for theory development in family firm research. *Journal of Family Business Strategy*, 5(1), pp.116-128. doi: <https://doi.org/10.1016/j.jfbs.2013.12.002>

- Thank you for your suggestions of SEM-models, this is very useful. We have now clarified in the manuscript that a detailed statistical plan will be created as part of the project's further development, including that partial least square SEM models will be considered (line 289-292):

“Our primary approach will be to use a causal mediation analysis framework, as suggested by Imai et al. (67), but a detailed statistical plan will be developed as the project proceeds and alternative approaches, such as partial least squares-SEMs (PLS-SEM) (68) will also be considered.”

5. Spatial overlay analysis (page 8, line 11), while the overlay would identify the area-based variables coinciding with temperature, I believe this visualization can be future developed based on spatial clustering methods such as LISA clustering or multivariate clustering. These are better for hotspot identification. Such methods will identify and visualize the spatial pattern of high-low clusters in temperature with other variables such as high-low UNEs and high deprived-low deprived demographic regions. I also think the authors could clarify a bit more on what type of overlay (e.g., identity, intersect) they like to use. Please check the link for LISA: https://geodacenter.github.io/workbook/6a_local_auto/lab6a.html

- Thanks for the suggestion. We have now included in the manuscript that alternative clustering approaches will be considered (line 307-309):

“We will assess the approach and consider different methods based on spatial clustering, such as LISA or multivariate clustering (69, 70).”

We have also addressed what type of overlay may be considered (line 313-315):

“We primarily aim to use a simple combination overlay analysis, but more complex methods, such as union, intersection, or identity operations may also be considered.”

6. Software development (Page 9, line 27): I am really excited to learn about this tool. I think the authors might want to discuss a little more how the results would be incorporated in GreenUr. Such as will there be an additional function to use the modeled relations from this study to estimate health

impact from LST raster maps? Will it be possible to draw dose-response function from GreenUr? What meteorological inputs would be required and which model (line 36)?

- GreenUr is under current development and the approach for incorporating heat data is not defined, but this is something that will be developed as part of the Green CURIOCITY project. The more specific methods will be determined by results currently not available. We have clarified in the text (line 328-330):

“GreenUr is a prototype under development and the exact input data and what functions to incorporate will depend on the general evolvement of Green CURIOCITY and on the current processing of the prototype.”

7. Testing different spatial units (page 7, line 8): It would be good to provide some details on the spatial units. How were these defined, and how can they be sensitive? Are these different buffer sizes, admin boundaries? As I found in the HELIX cohort paper indicated 100, 300, and 500 m buffer?

- Thank you, we have removed that part of the sentence to avoid confusion. We explain the buffer size approach on line 262-266:

“Greenness values are calculated and averaged within 100-, 300-, and 500-m buffers around each residential address. The values are averaged for buffers zones of 100, 300, and 500 m around residence. These buffer zones are commonly used in epidemiological studies assessing the impact of natural spaces on health outcomes. In general, the 300m buffer zone is considered as most indicative of home exposure (64), but additional buffer zones will also be tested for sensitivity analyses.”

As a whole, this is an exciting and much-needed longitudinal study! I am looking forward to seeing the results of this study.

- Thank you!

VERSION 2 – REVIEW

REVIEWER	Fry, Richard Swansea University, Medical School
REVIEW RETURNED	26-Nov-2021

GENERAL COMMENTS	Thank you for the clarifications - I look forward to the results from the study.
--

REVIEWER	Liu, Tao Guangdong Provincial Institute of Public Health
REVIEW RETURNED	05-Nov-2021

GENERAL COMMENTS	Thanks for the authors' detailed response to my concerns. All my questions were satisfactorily answered, and the quality of this manuscript has been improved largely. I suggest to accept it for publication.
--

REVIEWER	Labib, SM Cambridge University, MRC Epidemiology unit
REVIEW RETURNED	30-Oct-2021

GENERAL COMMENTS	Thanks for addressing all my concerns. Looking forward to the interesting outcomes from the project.
--